# Mindfulness, Interoception, and Olfaction: A Network Approach

**DOI:** 10.3390/brainsci10120921

**Published:** 2020-11-29

**Authors:** Barbara Lefranc, Charles Martin-Krumm, Charlotte Aufauvre-Poupon, Benoit Berthail, Marion Trousselard

**Affiliations:** 1APEMAC/EPSAM, EA 4360, Ile du Saulcy, BP 30309, CEDEX 1, 57006 Metz, France; charles.martinkrumm@gmail.com (C.M.-K.); marion.trousselard@gmail.com (M.T.); 2French Armed Forces Biomedical Research Institute, BP73, CEDEX, 91223 Brétigny-sur-Orge, France; 3Ecole de Psychologues Praticiens, Institut Catholique de Paris (Catholic Institute of Paris), VCR/ICP EA 7403-23, rue du Montparnasse, 75006 Paris, France; 4École Camondo, 266 Boulevard Raspail, 75014 Paris, France; charlotte.poupon@gmail.com; 5French Military Health Service Academy, 1 Place Alphonse Laveran, CEDEX 05, 75230 Paris, France; benoitberthail@hotmail.fr; 6Réseau ABC des Psychotraumas, 34000 Montpellier, France

**Keywords:** causal network, clustering, interoception, mindfulness, olfaction

## Abstract

The fine-tuned interplay between the brain and the body underlies the adaptive ability to respond appropriately in the changing environment. Mindfulness Disposition (MD) has been associated with efficient emotional functioning because of a better ability to feel engaged by information from the body and to notice subtle changes. This interoceptive ability is considered to shape the ability to respond to external stimuli, especially olfaction. However, few studies have evaluated the relationships between interoception and exteroception according to MD. We conducted an exploratory study among 76 healthy subjects for first investigating whether MD is associated with better exteroception and second for describing the causal interactions network between mindfulness, interoception, emotion, and subjective and objective olfaction assessments. Results found that a high level of MD defined by clustering exhibited best scores in positive emotions, interoception, and extra sensors’ acuity. The causal network approach showed that the interactions between the interoception subscales differed according to the MD profiles. Moreover, interoception awareness is strongly connected with both the MD and the hedonic value of odors. Then, differences according to MD might provide arguments for a more mindful attention style toward interoceptive cues in relation to available exteroceptive information. This interaction might underlie positive health.

## 1. Introduction

We constantly act upon the world while our brain continuously integrates information from inside and outside the body. Through the body, the brain receives information about the state of the external world (exteroception) and the body’s physiological state (interoception) [1]. The fine-tuned interplay between the brain and the body underlies the adaptive ability to respond appropriately in a constantly changing environment. It refers to an efficient body awareness which is the individual’s ability to feel engaged by information from the body and to notice subtle changes [2].

Furthermore, the construction of body awareness refers to a particular kind of mindful, non-judgmental awareness and a sense of self, grounded in physical external as internal sensations in the present moment [3]. Such an assertion contradicts the usual conceptualization in the medical and psychological literature. Body awareness has been mainly investigated in studies of psychiatric disorders as a cognitive attitude characterized by “an exaggerated patient focus on physical symptoms, magnification (“somatosensory amplification”), rumination, and beliefs of catastrophic outcomes” [2]. However, accumulated evidence supports the clinical benefits of body awareness [4]. Findings from numerous studies suggest the interest of improving body awareness in the management of chronic diseases including somatic [5,6,7] and psychiatric chronic diseases [8,9], as of chronic pain [10].

Thus, one of the challenges for health is to identify individual psychological resources associated with an efficient body awareness that can be developed. One pertinent candidate could be the Mindfulness Disposition (MD), which characterizes the awareness that emerges by paying attention on purpose, in the present moment, and non-judgmentally to the unfolding experience that is moment by moment [11,12]. Whether MD has been conceptualized as a stable ability to be mindful in everyday life, regardless of events [13], it is developed by mindfulness intervention [11,14]. This disposition has been associated with various positive physical and psychological health and with protective functioning (e.g., positive psychology, improvement of the quality of life) for both healthy subjects and patients [14,15,16]. More precisely, MD has been associated with resilience and its maintenance through attentional abilities, psychological flexibility, and efficient emotional regulation [16,17] that contribute to reduced perceived stress [18,19]. Furthermore, MD has been proposed to be associated with effective interoception. That is an efficient perception of the body from the inside that contributes to the regulation of physiological integrity (homeostasis) and the associated affective feelings, drives, and emotions. On one hand, MD has been positively related to a high level of interoceptive sensibility as assessed with the Multidimensional Assessment of Interoceptive Awareness questionnaire (MAIA) [3,20,21,22,23]. On the other hand, however, some work has reported mixed evidence of the association between mindfulness and interoception accuracy (mostly in using the heartbeat tracking paradigm) [24,25,26,27,28,29]. Overall, it has been proposed that MD is positively linked to interoceptive awareness [23]. This positive association is implicated in the benefits of MD in emotion regulation and cognition [17].

However, it is curious to note that no studies have investigated exteroception (i.e., the perception of surrounding stimuli through the classical sensory organs) in the MD. Yet, we are aware that the integration of multimodal exteroceptive signals (e.g., vision, sound, touch, olfaction, taste), vestibular and proprioceptive systems, and voluntary motor systems contribute to the exteroceptive body awareness (or “body schema”). Moreover, dysfunctions in exteroception impact patients’ body experience and functionality [30,31,32,33]. This implicit knowledge that we have of our body in relation to space and movement is in fact related to interoceptive body awareness [34]. Together, the interoception and exteroception body awareness emphasize the internal representation we have of our body and posture in relation to the environment [35]. Whether they have an impact on each other to guide action, the interoceptive ability is considered to shape the ability to respond to external stimuli over the long term [36].

This interaction is highlighted by studies using neuroimagery, which have shown the key role of the insular cortex in the integration of multimodal information and in interoceptive and exteroceptive processing [36]. Then, studies in chemosensory domains show that the central dorsal insula presents a chemosensory overlap of gustatory and olfactory sensations [37]. With respect to olfaction, a negative relationship between cardiac interoceptive accuracy and olfactory functioning (assessed by detection threshold, odor discrimination, and odor identification) has also been found [38]. Olfaction is an integral part of daily life, from deciding what to eat [39] to recalling autobiographical memories [40], or even choosing a partner [41]. As one of our five senses (i.e., sight, hearing, smell, taste, touch), it helps interpret the salience of surrounding external stimuli and determines much of the human experience. Olfactory research reveals important relationships between the sense of smell and many aspects of the psychological process involving human adaptation [42]. First, many studies suggest that odors can modulate mood, cognition, and behavior in healthy subjects [43]. Furthermore, stress can affect the olfactory detection threshold of a malodor, suggesting a close relationship between stress-related sensory hypervigilance and the olfactory system [44]. Second, this close relationship between olfactory and affective information processing is highlighted both by the incidence of major depression in anosmic subjects [45] and the importance of olfactory disorders in the case of mood disorders [46].

Overall, these findings suggest a situational interaction of intero- and exteroceptive stimulus processing that may depend on MD and extend to emotional states. However, to our knowledge, few studies have evaluated the relationships between interoception, exteroception, and emotion in healthy subjects, and even fewer their causal interactions. Verdonk et al. [17] reviewed the Event-Related Potential (ERP) literature related to mindfulness for a better understand of how mindfulness works. The review suggests that mindfulness would facilitate the conscious processing of information that comes from inside and outside the body. Unfortunately, the existing ERP literature does not allow for hypotheses to be formulated on the causal interaction network between mindfulness, interoception, exteroception, and emotion. There are several models for studying causal interactions. A causal interaction model is a set of mechanisms, a set of causes, and an effect. It relaxes the restrictions of causal independence models in which each cause has a unique mechanism variable, and each mechanism variable has a unique cause [47]. For example, a causal interaction model is relevant for conceptualizing a mental disorder as resulting from causal interactions network between symptoms, rather than as an underlying pathological entity [48]. Psychiatric symptoms have been argued as reciprocal rather than common cause effects. Then, disorders are studied using a network approach to evaluate the network of symptoms (“nodes”) presumed to be causal and the connections between them (“edges”) [48]. Such a causal interaction model could provide a way to investigate the network of causal relations between interoception, exteroception, emotion, and mindfulness, functioning as systems of interacting variables.

In this paper, we aim to evaluate the relationship between MD, interoception, exteroception, and emotion using three questions. The first issue is concerned with the causal interactions network between mindfulness, interoception, exteroception, and emotion. We make the hypothesis that reciprocal relationships exist between them. The second issue will part in a categorical approach focusing on whether the MD is associated with differential interoception, exteroception, and emotion. Our hypothesis is that subjects with MD exhibited a higher level of interoception, exteroception including olfaction and emotion. The third aims to evaluate how MD interacts with interoception, olfaction, and emotion using a causal network approach. We assume that the network of causal relations differs according to the MD profile, with more connections among interoception, olfaction, and positive emotion for high MD groups.

## 2. Material and Methods

### 2.1. Participants and Design

The sample included 76 civilians and military scheduled for a marine, submarine, or polar overwintering a few weeks later. They took part in an exploratory pragmatic study aiming to evaluate the impact of isolated and confined environments on human adaptation. The baseline was realized during the autumn season of 2017–2018 for the sailors and during the autumn season of 2018–2019 for the polar winterers. See Table 1 for demographic information. The follow-up of the cohort is yet in progress.

The study has been approved by the Comité de Protection des Personnes sud-est VI (France) in September 2017 (ID-RCB: 2017-A01329-44). After a complete description of the study, written informed consent for participation in this low-risk study was obtained.

### 2.2. Measures

The collected socio-demographic included: age, gender, marital status, submarine experience, position occupied in the Sub-Surface Ballistic Nuclear (SSBN).

The auto-questionnaire used to assess mindful status was the Freiburg Mindfulness Inventory (FMI, 14 items) (Appendix A). It measures dispositional trait mindfulness by indexing facets of Presence and Non-judgmental acceptance. It is semantically independent of a meditation context and it is applicable to all population groups, in particular those with no practice of mindfulness meditation. It is scored using a four-point scale, with responses ranging from 1 (rarely) to 4 (almost always). A total mindfulness score was computed by summing items except for the 13th item, which was reversed [49,50].

Interoceptive awareness was evaluated using the 32-item MAIA questionnaire (multidimensional assessment of interoceptive awareness) (Appendix A) [3] that measures eight facets: (i) Noticing (awareness of uncomfortable, comfortable, and neutral body sensations), (ii) Not-Distracting (tendency to ignore or distract oneself from sensations of pain or discomfort), (iii) Not-Worrying (emotional distress or worry with sensations of pain or discomfort), (iv) Attention Regulation (ability to sustain and control attention to body sensation), (v) Emotional Awareness (Awareness of the connection between body sensations and emotional states), (vi) Self-Regulation (ability to regulate psychological distress by attention to body sensations), (vii) Body Listening (actively listens to the body for insight), and (viii) Trusting (experiences one’s body as safe and trustworthy). This questionnaire considers the adaptive aspects of body awareness (i.e., as a present-moment and attention style to body sensations), which contrast with anxiety-driven hypervigilance to body sensations [2]. It is scored using a six-point scale, with 5 responses ranging from 0 (never) to 5 (always).

Emotional functioning was assessed using the scale of positive and negative emotional experiences (Scale of Positive and Negative Experience, SPANE) (Appendix A) [51,52], including 12 items to be rated by the subject on a five-point Likert scale (1 = very rarely or practically never, to 5 = very often or always). The latest passed week was considered.

For exteroception, we used a homemade questionnaire (Appendix A) including a ten-point Likert Scale (0 = low, 10 = high). It assessed the subjective exteroceptive acuity for each of the exterosensors: vision, sound, touch, olfaction, taste, and equilibrium.

Control of the nasal functioning has been realized using the DyNaChron (Dysfonctionnement Nasal Chronique) autoquestionnaire. This questionnaire is usually used in the case of chronic nasal dysfunction [53]. It assesses the olfactory and gustatory discomfort of the subjects on a daily basis.

The smell test ETOC (European Test of Olfactory Capabilities) [54] was used to assess the olfactory sensitivity of individuals. Individuals must smell the contents of the test tubes (16 sets of 4 test tubes) to find the one containing an odor (discrimination) and the nature of this same odor (identification). An evaluation of the hedonic value of the detected odor is added. This task was only applied for the subsample of 31 subjects of the cohort.

### 2.3. Data Analysis

Data analyses were performed using Python (Python, Software Foundation, v3.8, Wilmington, DE, USA) and RStudio (RStudio, v1.3.1093, Boston, MA, USA). All scales demonstrated acceptable levels of internal consistency in our sample and subsample (Cronbach’s between 0.75 and 0.85). First, a k-means, unsupervised machine learning algorithm using the scikit-learn library in Python, was applied to categorize MD groups according to FMI Presence and Non-judgmental acceptance subscales for all subjects. Second, k-means was applied to categorize MD groups according to FMI Presence and Non-judgmental acceptance subscales only for subjects that complete the ETOC. Due to operational preparations for the mission, not all participants were able to complete the ETOC. Whatever the sample, the following methodology used is identical. FMI data of participants were previously standardized by removing the mean and scaling to unit variance using the StandardScaler function. Statistical analyses were used to assess the impact of MD clustering on olfactory sensitivity. The Shapiro-Wilk test was used to determine whether data were normally distributed. Homoscedasticity was assessed using the Brown-Forsythe Levene type test. The statistics were adopted following the results of the previous tests. *t*-test or nonparametric Mann-Whitney analyses were performed individually to explore the presence of significant differences in the olfactory sensitivity according to the MD groups. For all analyses, statistical significance was set at *p* < 0.05. Trends to a difference were considered when 0.05 < *p* < 0.11. For significant analyses, the effect size and their confidence intervals (CI) were reported.

Interaction networks (*n* = 4) were estimated separately for all the subjects and those distributed between both MD groups to explore the connections between mindfulness, interoception, emotions, and subjective exterosensors acuity. Another interaction network was estimated for participants who completed the ETOC using RStudio. A set of variables was considered for all subjects including interoception subscales, mindfulness subscales, positive and negative emotions, and subjective olfactory acuity. For those who completed the ETOC test, the set of variables included interoception and mindfulness subscales, subjective olfactory acuity, and the ETOC performances. Whatever the participants, the methodology used to develop interaction networks is identical. Gaussian graphical models based on partial Pearson correlations were used to evaluate the networks among subjects using the qgraph package [55]. Then, the Least Absolute Shrinkage and Selection Operator (LASSO) regularization method was applied to reduce the likelihood of spurious edges. The Group LASSO (GLASSO) was reported in figures. An Extended Bayesian Information Criterion (EBIC) model was introduced and fixed in the LASSO. Positive edges are shown in green, negative edges are in red, and the relative strength of the edge weight is reflected by edge thickness. Centrality indices were calculated to assess the centrality of the variables related to the strength of its connections with the other variables in the network. The evaluation of the accuracy and stability of the estimated network was determined by bootstrapping using the bootnet package [55]. Nonparametric bootstrap was estimated for edge weights. The stability of strength centrality values was evaluated using the case-dropping subset bootstrap and estimating correlations between the original centrality index and the values obtained from subsets of the data. Plots of edge weight accuracy and stability of strength centrality are in the Appendix A. Within this network analysis framework estimated for all the subjects and those distributed between both MD profiles, interoception subscales, mindfulness subscales, and subjective olfactory acuity were represented as nodes, and connections between symptoms were undirected edges. Edge weights (strength of connections between two symptoms) correspond to partial correlations that control for all other variables in the network. The overall importance of each variable was evaluated in terms of node centrality strength, which is the sum of absolute values of all edge weights connected to a node. Another interaction network was estimated for participants who completed the ETOC test without clustering using the same method and R packages as before. Within this network analysis framework, interoception subscales, mindfulness subscales, subjective olfactory acuity, and ETOC performances were represented as nodes, and connections between symptoms were undirected edges.

## 3. Results

Causal interaction network between mindfulness, interoception, emotions, and subjective exterosensors acuityA causal interaction network was modelized on all the subjects (*n* = 76) to explore the connections between mindfulness, interoception, emotions, and subjective extrasensors acuity. The network is shown in Figure 1.

The most important connection is between interoception subscales. More precisely, there are notable connections between noticing and emotional awareness. Important connections are also highlighted between self-regulation and body listening, noticing and body listening, presence and non-judgmental acceptation, attention regulation, and non-judgmental acceptation. Smaller connections take place between subjective olfactory acuity and subjective taste acuity, presence and self-regulation, attention regulation and self-regulation, trusting and non-judgmental acceptation, attention regulation and body listening, emotional awareness and body listening, and noticing and body listening. Weaker connections are found between subjective sight acuity and subjective equilibrium acuity and positive emotions and attention regulation.

The CS-coefficient for edge weights was (CS(cor = 0.7) = 0.21), and the CS-coefficient for strength centrality was (CS(cor = 0.7) = 0.21), indicating a sufficient level of stability for interpreting rank order of edge weights and strength centrality. Bootstrapping CIs are used to interpret network connections (Figure 2A). The figure reveals sizable bootstrapped CIs around the estimated edge-weights, indicating that many of the edge-weights likely do not significantly differ from one another. The LASSO regularization was also used in the case of partial correlation coefficients for indicating which connections are strong enough to be included in the network.

The most central nodes that emerge from the centrality table are linked to interoceptive subscales, including body listening, attention-regulation, self-regulation, and non-judgmental acceptation (Figure 2B). Most notably, body listening was one of the most central variables in the estimated network (Appendix A for difference tests of node strenght and centralities between all pairs of edge-weight).

### 3.1. Causal Interaction Network between Interoception, Emotions, and Subjective Olfactory Acuity

A causal interaction network was modelized on all the subjects without group distinction (*n* = 76) to explore the connections between interoception, emotions, and subjective olfactory acuity. The network is shown in Figure 3.

The most important connection is between interoception subscales. More precisely, there are notable connections between noticing and emotional awareness. Important connections are also highlighted between self-regulation and body listening. Smaller connections take place between noticing and body listening, emotional awareness and body listening, attention regulation and body listening, self-regulation and attention regulation, positive emotions and attention regulation, trusting and positive emotions, negative emotions and positive emotions, not distracting and body listening, and negative emotions and self-regulation. Weaker connections are found between attention regulation and emotional awareness, not distracting and attention regulation, self-regulation and trusting, attention regulation and trusting, subjective olfactory acuity and trusting, positive emotions and subjective olfactory acuity, attention regulation and subjective olfactory acuity, and body listening and subjective olfactory acuity.

The CS-coefficient for edge weights was (*CS*(cor = 0.7) = 0.43), and the CS-coefficient for strength centrality was (*CS*(cor = 0.7) = 0.43), indicating a good level of stability for interpreting rank order of edge weights and strength centrality. Bootstrapping CIs are used to interpret network connections (Figure 4A). The figure reveals linear bootstrapped CIs around the estimated edge-weights, indicating that many of the edge-weights likely significantly differ from one another. The LASSO regularization was also used in the case of partial correlation coefficients for indicating which connections are strong enough to be included in the network.

The most central nodes that emerge from the centrality table are linked to interoceptive subscales, including body listening, attention-regulation, emotional awareness, and self-regulation (Figure 4B). Most notably, body listening was one of the most central variables in the estimated network (Appendix A for difference tests of node strenght and centralities between all pairs of edge-weight).

### 3.2. Impact of MD Clustering on Psychological Assessments

The k-means characterizes two groups (high vs. low mindful) based on the FMI score of all the participants (Figure 5).

The high mindfulness (*n* = 32) profile exhibits a higher MD by showing high scores on FMI Presence and Non-judgmental acceptance subscales. The low mindfulness (*n* = 44) profile exhibits a lower MD by showing lower scores on Presence and Acceptation FMI subscales. A summary of results according to MD profiles on psychological assessments is shown in Table 2.

In psychological functioning, there is a significant group effect for both SPANE positive (W = 483, *p* = 0.01, r = 0.27, 95% CI [0.06,0.47]) and negative (t = 2.23, *df* = 72.49, *p* = 0.02, d = 0.51, 95% CI [0.04,0.98]) score. The high mindfulness group has higher positive emotions than the low mindfulness group and the opposite for the low mindfulness group.

In interoceptive functioning (MAIA), we only found differences for five subscore dimensions. There are significant group effects for attention regulation (t = −5.15, *df* = 70.58, *p* = 0.000, d = 1.17, 95% CI [0.68,1.68]), self-regulation (t = −3.88, 68.99, *p* = 0.000, d = 0.90, 95% CI [0.41,1.38]), body listening (W = 446.5, *p* = 0.006, r = 0.31, 95% CI [0.09,0.51]), and trusting (W = 349, *p* = 0.000, r = 0.43, 95% CI [0.22,0.60]). Whatever the dimension, the high mindfulness group has a higher score than the low mindfulness group. There is also a tendency to a group effect (t = 1.84, *df* = 66.25, *p* = 0.06, d = 0.43, 95% CI [0.04,0.90]) for not distracting. The low mindfulness group is less distracted than the high mindfulness group.

In subjective exteroceptive acuity, we found differences for each of the exterosensors except for touch. There are significant group effects for olfaction (W = 455, *p* = 0.008, r = 0.30, 95% CI [0.09,0.49]), taste (W = 500, *p* = 0.02, r = 0.25, 95% CI [0.05,0.46]), hearing (W = 446, *p* = 0.005, r = 0.32, 95% CI [0.12,0.51]), and equilibrium (W = 494, *p* = 0.02, r = 0.26, 95% CI [0.05,0.47]). Whatever the exterosensors, the high mindfulness group has a higher subjective acuity than the low mindfulness group. There is also a tendency to group effect for vision (W = 551, *p* = 0.08, r = 0.20, 95% CI [0.02,0.41]). The high mindfulness group tends to have higher subjective acuity of vision than the low mindfulness group.

### 3.3. Impact of MD Clustering on Olfactory Sensitivity Assessments

The k-means characterizes two groups (high vs. low mindful) based on the FMI score of the participants that complete the ETOC (Figure 6).

The high mindfulness (*n* = 18) profile exhibits a higher MD by showing high scores on FMI Presence and Non-judgmental acceptance subscales. The low mindfulness (*n* = 13) profile exhibits a lower MD by showing lower scores on Presence and Acceptation FMI subscales. A summary of results according to MD profiles on psychological assessments is shown in Table 3.

The dynachron, a test controlling the functionality of the nose, has been assessed and indicated normal values for all study subjects. In olfaction sensory, there is a tendency to a group effect for identification of the odor (W = 77.5, *p* = 0.106, r = 0.30, 95% CI [0.08,0.61]). The high mindfulness group tends to have a better ability to identify odors than the low mindfulness group. The impact of MD on causal interaction network is shown.

Two causal interaction networks were modelized according to the MD groups (high mindfulness vs. low mindfulness). The network for both MD groups is shown in Figure 7.

In low MD (*n* = 44; Figure 7A), the connections are between interoception subscales. The most important connection is mainly between emotional awareness and noticing. Other connections are between emotional awareness and body listening, and between body listening and noticing. In the high MD (*n* = 32; Figure 7B the most important connection is between interoception subscales including self-regulation and body listening. Important connections are also highlighted between body listening and auto-regulation, between emotional awareness and noticing, and between autoregulation and positive emotions.

The Correlation Stability (CS) coefficient for edge weight for low (*CS*(cor = 0.7) = 0.14) and high (*CS*(cor = 0.7) = 0.22) MD networks, and the CS-coefficient for strength centrality weight for low (*CS*(cor = 0.7) = 0.14) and high (*CS*(cor = 0.7) = 0.28) indicate a sufficient level of stability to evaluate rank order of edge weight and strength centrality.

For the low MD group, bootstrapping Cis are used to interpret network connections (Figure 8A). The CI, estimating the strength of connection, of most interactions is relatively large, suggesting that replication with other samples is necessary to elucidate the strength of these interactions. However, in the case of partial correlation coefficients, the LASSO regularization aims at estimating the connections that do not necessarily have to be exactly zero. Consequently, observing that a connection is not set to zero already indicates that it is strong enough to be included in the network.

The most central nodes that emerge from the centrality table are interoception subscales including emotional awareness, noticing, and body listening for the low MD group (Figure 9A). Most notably, emotional awareness was one of the most central variables, closely followed by noticing in the estimated network (Appendix A for difference tests of node strenght and centralities between all pairs of edge-weight).

For the high MD group, bootstrapping CIs are used to interpret network connections (Figure 8B). The LASSO regularization was also used in the case of partial correlation coefficients for indicating which connections are strong enough to be included in the network.

The most central nodes that emerge from the centrality table are body listening, self-regulation, emotional awareness, and attention regulation for the high MD group (Figure 9B). Most notably, body listening was one of the most central variables in the estimated network (Appendix A for difference tests of node strenght and centralities between all pairs of edge-weight).

### 3.4. Causal Interaction Network Using Objective and Subjective Olfaction Data

A causal interaction network was modelized on the subjects who completed the ETOC test (*n* = 31). The network is shown in Figure 10.

The most important connection is between interoception subscales, FMI Presence, and Non-judgmental acceptance subscales and the ETOC test. More precisely, there are notable connections between noticing and emotional awareness, presence and non-judgmental acceptance, body listening and hedonic value of the ETOC test, body listening and self-regulation, and presence non-judgmental acceptance and attention regulation. Important connections are also highlighted between attention regulation and self-regulation, identification of ETOC odors and subjective olfactory acuity, body listening, and attention regulation. Smaller connections take place between presence and body listening, non-judgmental acceptance and body listening, trusting and non-judgmental acceptance, and self-regulation and trusting. Weaker connections take place between presence and self-regulation, attention regulation and presence, and detection of ETOC odors and subjective olfactory acuity.

The CS-coefficient for edge weights was (*CS*(cor = 0.7) = 0.07), and the CS-coefficient for strength centrality was (*CS*(cor = 0.7) = 0.07), indicating a level of stability to be interpreted with caution for interpreting rank order of edge weights and strength centrality. Bootstrapping CIs are used to interpret network connections (Figure 11A). The figure reveals sizable bootstrapped CIs around the estimated edge-weights, indicating that many of the edge-weights likely do not significantly differ from one another. The LASSO regularization was also used in the case of partial correlation coefficients for indicating which connections are strong enough to be included in the network.

The most central nodes that emerge from the centrality table are body listening subscale, non-judgmental acceptance subscale, autoregulation, and self-regulation (Figure 11B). Most notably, body listening was one of the most central variables in the estimated network (Appendix A for difference tests of node strenght and centralities between all pairs of edge-weight).

## 4. Discussion

The first issue is concerned with the causal interaction network between mindfulness, interoception, exteroception, and emotion. The first hypothesis that reciprocal relationships exist between them is not confirmed. Nevertheless, the first network highlights a causal interaction network between the subscales of interoception, between mindfulness and interoception, between mindfulness and positive emotions, and between subjective olfactory and gustatory acuity. The first causal interaction network clarifies existing results on interoception conceptualized as a multidimensional construct with four key dimensions [3,56]. It shows that the last levels in interoception integration, which is the attentional, the trust, and the mind-body levels, are connected without connection with the first perceived body sensations level. Interestingly, mindfulness level is connected with the three last levels of interoception integration as with subjective olfactory and gustatory acuity. Although these connections between subjective evaluations are small, they are in concordance with prior findings proposing a neurofunctional explanatory model of how olfactory and gustatory exteroceptions trigger the integration of intero- and exteroceptive sensations [36]. Finally, when mindfulness is removed and exteroception reduced to subjective olfactory acuity, results highlight higher connections between the subjective evaluations of the three last levels of interoception, olfactory acuity, and positive emotions. This result asks about the impact of mindfulness level in the observed connections.

The categorical approach aims to evaluate the relationship between MD, interoception, exteroception, and emotion. First, we observed that a high level of MD (i.e., awareness that emerges by paying attention on purpose, in the present moment, and non-judgmentally to the unfolding experience that is moment by moment), as defined by the clustering method, exhibited a more positive emotional state with less negative emotions and more positive emotions. In accordance with the literature on the relationship between mindfulness and emotional functioning [57,58], these results confirm that the two groups are correctly identified and relevant for studying how interoceptive and exteroceptive might interact in the underlying mindfulness mechanisms.

Except for not distracting interoception subscale, the high MD profile was characterized by a higher level of interoception in terms of ability to sustain and control attention to body sensation (attention regulation subscale), to regulate psychological distress by attention to body sensations (self-regulation subscale), to be aware of the connection between body sensations and emotional states (body listening subscale), and to experience one’s body as safe and trustworthy (trusting subscale). The high level of not distracting observed for the low MD subjects compared to the high MD subjects might reflect a tendency to ignore oneself from sensations of pain or discomfort that should arrive few weeks before the isolated and confined mission. Interestingly, the high MD subjects exhibited a higher subjective acuity for each of the exterosensors (except the touch) with only a tendency to a difference between the two MD profiles for the vision. The higher subjective acuity for audition, olfaction, and taste for the high MD group is in accordance with both the neurophysiological data highlighting the role of the insula in sensory information integration [36,59] and the better postero- and antero insula functioning after mindfulness training [60]. To date, there is much speculation about insula function in equilibrium, although preclinical evidence exists that the somatosensory and proprioceptive information more strongly activate the posterior insula cortex as compared to the anterior one [61]. In addition, in the subsample of participants with the ETOC performances, the objective evaluation of olfaction tends to differ between both MD profiles only for the identification performances. This lack of results might be due to either a high level of performance in the subsample with very few errors in both olfactory tasks, or a too small size of the sample, or both. Finally, the higher self-report of exteroceptive acuity might indicate that high MD subjects are characterized by either better attention on exteroceptive information, or more confidence in this external information, or a mix of them. The neural exteroceptive processing and its relationship with interoceptive abilities that participate in body awareness must be further studied using neurophysiological investigations for understanding the insula’s role as a neural center for the establishment of the psychological construct of the embodied body in humans.

Whether the present results suggest that interoception and olfactory exteroception interact differently according to MD, they need to be completed by the causal network approach. In accordance with our second aim, our main hypothesis is that causal network relationships differ according to the MD profile, with more connections among interoception, olfaction, and emotion for the high MD group only partially validated. What we have here with the causal interaction network approach is that the interoception subscales interactions differed according to the MD profiles. More precisely, we observed an interaction between the interoception subscales for the low MD, contrary to larger connexions in the estimated network for the high MD. For the high MD, there are strong interactions between interoception subscales and positive emotions. These results were supported by the second interaction network that considers all the subjects. The major findings are interactions between interoception and both positive and negative emotions. These results are in line with the interoception to pay attention to body sensation and thus linked to better emotion regulation [58]. Effective emotion regulation involves the ability to evaluate with accuracy the physiological reactions to events. Moreover, the second interaction network highlights evidence for connections between interoception, positive emotions, and subjective olfactory acuity. Soudry et al. [62] in a review claim common brain structures to emotions and olfaction process, olfaction as emotion attributing positive and negative valence to the environment. Among these structures, we note the amygdala, the hippocampus, the insular cortex, and the orbitofrontal cortex. The relationship between the valence of emotion and odor was recently validated by Toet et al. [63] in healthy subjects.

Furthermore, for the sub-cohort, we did not validate the assumptions that mindfulness is most central and has the greatest spreading influence on interoception, olfaction, and emotion. Rather, we found that interoception awareness is strongly connected with both the MD and the hedonic value of odors. These results highlight the centrality of the body-listening subscale, the ability to actively listen the body for insight. We also found this centrality in individuals with a high MD. The interoception awareness seems to have a main role in the MD as well as in the ability to smell odors.

To our knowledge, this is the first study aiming to explore how mindfulness interact with information about the state of the external world (exteroception) and the body’s physiological state (interoception) using a causal interaction network. Results first highlight the relevance of the MD characterization for the studies underlying psychological and neurobiological mechanisms involved in mindfulness. Second, they suggest that the fine-tuned interplay between the brain and the body underlies emotional abilities to respond appropriately in a constantly changing environment. It refers to an efficient body awareness that might reflect the individual’s ability to feel engaged by information from the body and to notice subtle changes [2]. The differences according to MD might provide arguments for a more mindful attention style toward interoceptive cues in relation to the available exteroceptive information for underlying body awareness abilities. This attention style might be implied in the hedonic value toward exteroceptive information. Thirdly, studying exteroceptive abilities and their changes according to the environmental challenges need further investigation. Especially, olfactory sense might be considered as a singular sense in the ability to cope with adaptation. This question is of interest at least in the field of the isolated and confined environment which exposes the personals to cognitive, psychological, including depressive mood, and sensory disturbances [64,65]. The exceptional nature of these terrains according to the available exteroceptive information means that individuals participating in such missions must experiment and implement several processes to adapt. How exteroception abilities cope with the environmental challenge might be useful for monitoring personals’ health. Finally, a better understanding of how the body and the brain interplay may help develop countermeasures to protect workers’ health, particularly in situations where they are exposed to high environmental constraints and cognitively demanding situations (e.g., space flights, military operations, etc.).

## 5. Limitations

The exploratory study has two important shortcomings related to participants and materials. The first one refers to the studied population (i.e., mostly male, young, and selected subjects). The most important limitation is a small size. In these types of environments, it is a huge difficulty to consider due to time constraints, data collected in the ecological environment, and the impact of the environment itself on the psychosensory degradation of these professionals. Effective size is a well-known limitation for causal interaction network studies. Nevertheless, we have considered this limit using bootstrapping. Whether the characteristics of our sample limits a generalization of our results, namely, for women, they offer a very homogenous healthy population. The second one comes from the self-report measures as there are limitations inherent in the self-report approach to assessing any psychological dimensions that include, but are not limited to, response bias, state dependencies, and social desirability. Objective evaluation for exterosensors needs to be developed for healthy subjects with an evaluation of sensitivity for helping researchers to better investigate the role of exteroception in human adaptation. Furthermore, the version of the MAIA developed in 2012 has some problems that are addressed in Mehling et al. [56]. The new version of the MAIA (MAIA-2) includes a much broader awareness focus on thoughts and exteroceptive stimuli. This new scale might be more informative for our question.

## 6. Conclusions

This exploratory study explores how mindfulness interact with information about the state of the external world (exteroception) and the body’s physiological state (interoception) using interaction networks. First, results highlight interactions between mindfulness, interoception, emotions, and subjective olfactory acuity, underlying common brain structures in the literature. Secondly, they claim the relevance of the MD characterization for the studies underlying psychological and neurobiological mechanisms involved in mindfulness. Thirdly, they suggest that the fine-tuned interplay between the brain and the body underlies emotional abilities to respond appropriately in a constantly changing environment. Further investigation is needed to study exteroceptive abilities and their modifications function of the environmental challenges. A better understanding of how the body and the brain interplay may help develop countermeasures to protect workers’ health, particularly in situations where they are exposed to high environmental constraints and cognitively demanding situations (e.g., space flights, military operations, etc.).

## Figures and Tables

**Figure 1 brainsci-10-00921-f001:**
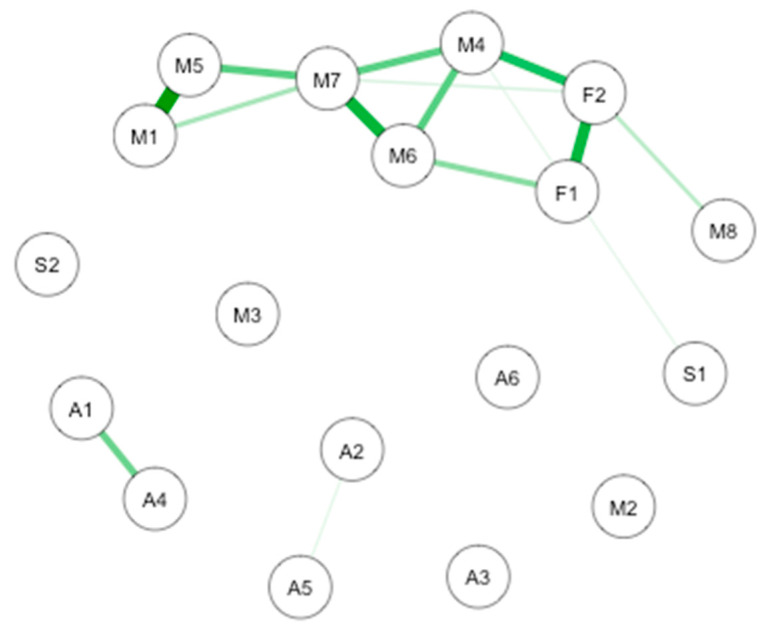
GLASSO-regularized partial Pearson correlation networks of interoception subscales, mindfulness subscales, and subjective olfactory acuity, resulting from all the subjects. Green edges denote positive associations, red edges negative ones. F1: Freiburg Mindfulness Inventory (FMI) presence; F2: FMI Non-judgmental acceptation; S1: Scale of Positive and Negative Experience (SPANE) positive emotion; S2: SPANE negative emotion; F1: FMI presence; F2: FMI Non-judgmental acceptation; M1: Multidimensional Assessment of Interoceptive Awareness questionnaire (MAIA) noticing; M2: MAIA not distracting; M3: MAIA not worrying; M4: MAIA attention regulation; M5: MAIA emotional awareness; M6: MAIA self-regulation; M7: MAIA body listening; M8: MAIA trusting; A1: Subjective olfactory acuity; A2: Subjective sight acuity; A3: Subjective hearing acuity; A4: Subjective taste acuity; A5: Subjective equilibrium acuity.

**Figure 2 brainsci-10-00921-f002:**
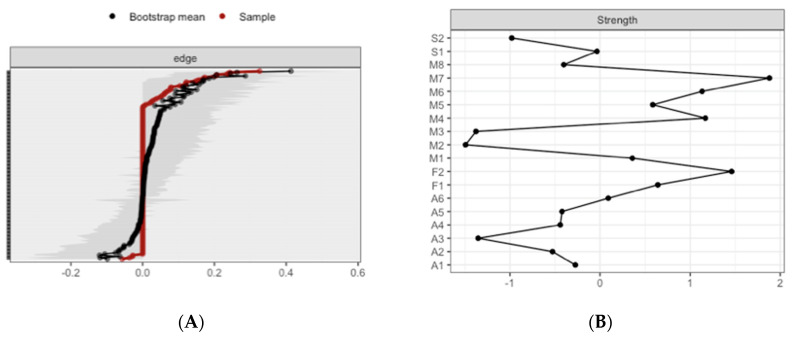
Estimated edge-weights for the estimated network and centrality of nodes. (**A**) Left: Bootstrapped CIs of estimated edge-weights for the estimated network. The red line indicates the sample values and the gray area the bootstrapped CIs. Each horizontal line represents one edge of the network, ordered from the edge with the highest edge-weight to the edge with the lowest edge-weight. (**B**) Right: Centrality of nodes within a network for all the subjects.

**Figure 3 brainsci-10-00921-f003:**
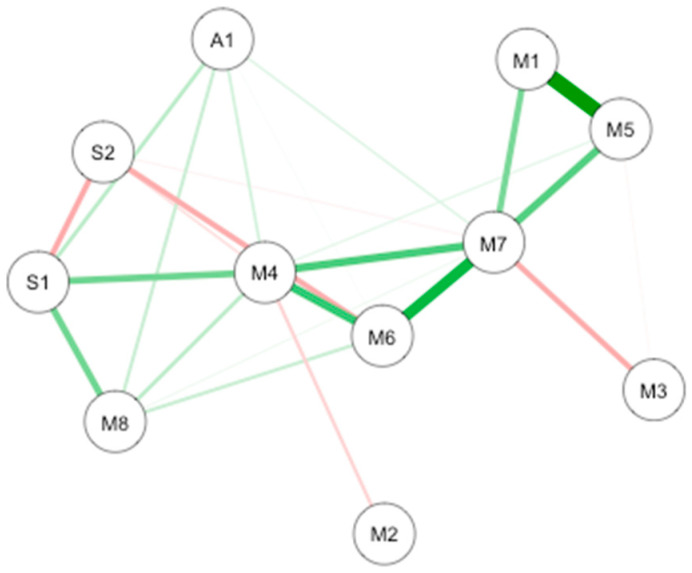
GLASSO-regularized partial Pearson correlation networks of interoception subscales, mindfulness subscales, and subjective olfactory acuity, resulting from all the subjects. Green edges denote positive associations, red edges negative ones. S1 SPANE positive emotion; S2: SPANE negative emotion; F1: FMI presence; F2: FMI Non-judgmental acceptation; M1: MAIA noticing; M2: MAIA not distracting; M3: MAIA not worrying; M4: MAIA attention regulation; M5: MAIA emotional awareness; M6: MAIA self-regulation; M7: MAIA body listening; M8: MAIA trusting; A1: Subjective olfactory acuity.

**Figure 4 brainsci-10-00921-f004:**
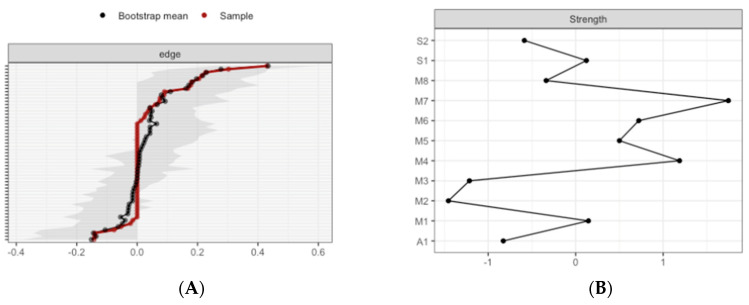
Estimated edge-weights for the estimated network and centrality of nodes. (**A**) Left: Bootstrapped CIs of estimated edge-weights for the estimated network. The red line indicates the sample values and the gray area the bootstrapped CIs. Each horizontal line represents one edge of the network, ordered from the edge with the highest edge-weight to the edge with the lowest edge-weight. (**B**) Right: Centrality of nodes within a network for all the subjects.

**Figure 5 brainsci-10-00921-f005:**
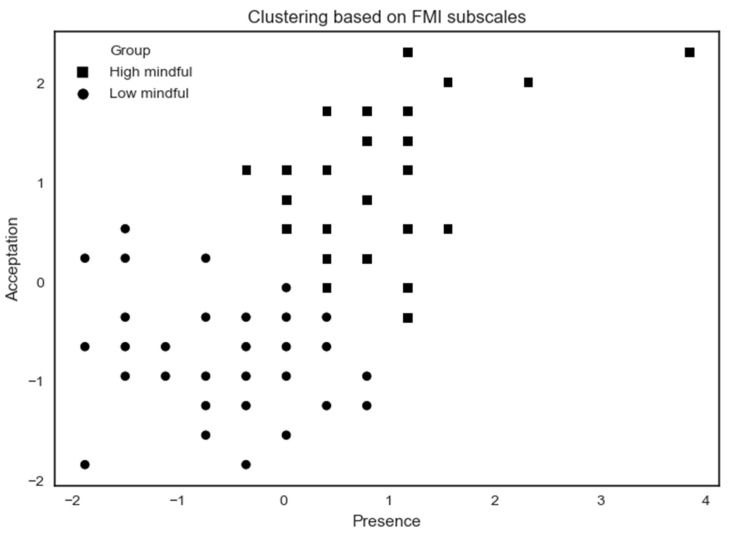
K-means clustering analysis applied on all subjects based on FMI subscales. Some participants share the same x- and y-coordinates causing them to merge at the same point.

**Figure 6 brainsci-10-00921-f006:**
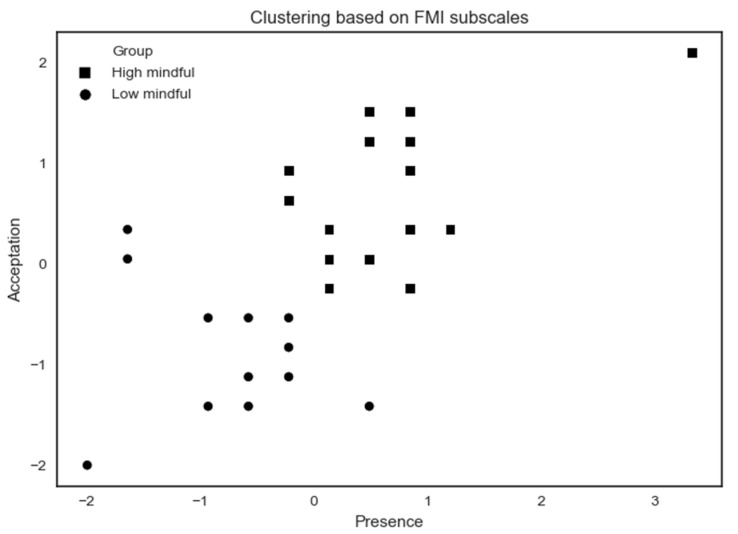
K-means clustering analysis applied to ETOC performance subjects based on FMI subscales. Some participants share the same x- and y-coordinates, causing them to merge at the same point.

**Figure 7 brainsci-10-00921-f007:**
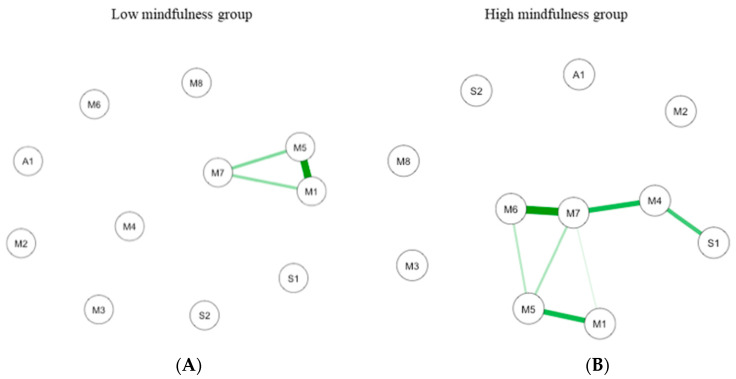
GLASSO-regularized partial Pearson correlation networks of interoception subscales, mindfulness subscales, and subjective olfactory acuity, resulting from low mindfulness (**A**) and high mindfulness groups (**B**). S1 SPANE positive emotion; S2: SPANE negative emotion; F1: FMI presence; F2: FMI Non-judgmental acceptation; M1: MAIA noticing; M2: MAIA not distracting; M3: MAIA not worrying; M4: MAIA attention regulation; M5: MAIA emotional awareness; M6: MAIA self-regulation; M7: MAIA body listening; M8: MAIA trusting; A1: Subjective olfactory acuity.

**Figure 8 brainsci-10-00921-f008:**
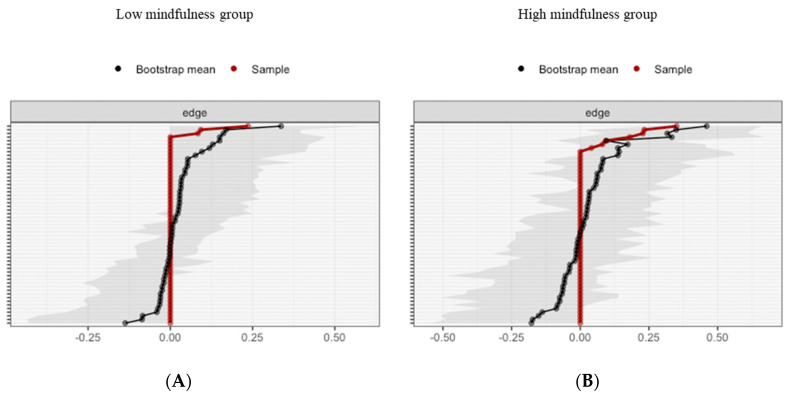
Estimated edge-weights for the estimated network and centrality of nodes. Bootstrapped CIs of estimated edge-weights for the estimated network for the low MD group (**A** left) and the high MD group (**B** right). The red line indicates the sample values and the gray area the bootstrapped CIs. Each horizontal line represents one edge of the network, ordered from the edge with the highest edge-weight to the edge with the lowest edge-weight.

**Figure 9 brainsci-10-00921-f009:**
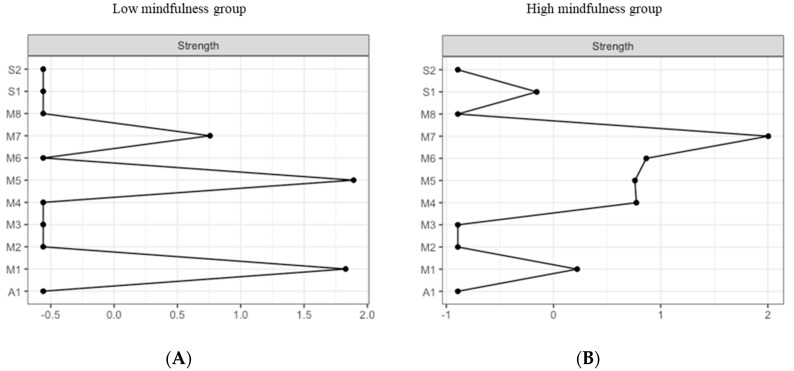
Estimated edge-weights for the estimated network and centrality of nodes. Centrality of nodes within a network according to low MD profile (**A** left) and the high MD profile (**B** right). S1: SPANE positive emotion; S2: SPANE negative emotion; F1: FMI presence; F2: FMI Non-judgmental acceptation; M1: MAIA noticing; M2: MAIA not distracting; M3: MAIA not worrying; M4: MAIA attention regulation; M5: MAIA emotional awareness; M6: MAIA self-regulation; M7: MAIA body listening; M8: MAIA trusting; A1: Subjective olfactory acuity.

**Figure 10 brainsci-10-00921-f010:**
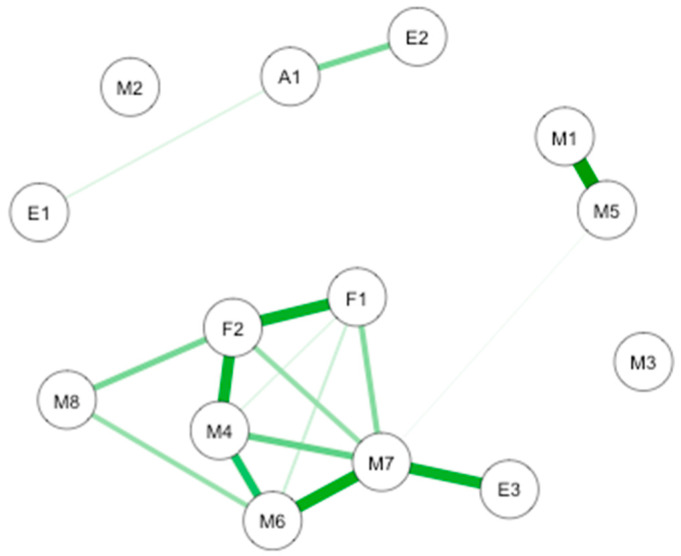
GLASSO-regularized partial Pearson correlation networks of interoception subscales, mindfulness subscales, emotional subscales, subjective olfactory acuity, and ETOC performances. F1: FMI presence; F2: FMI Non-judgmental acceptation; M1: MAIA noticing; M2: MAIA not distracting; M3: MAIA not worrying; M4: MAIA attention regulation; M5: MAIA emotional awareness; M6: MAIA self-regulation; M7: MAIA body listening; M8: MAIA trusting; A1: Subjective olfactory acuity; E1: ETOC detection; E2: ETOC identification; E3: ETOC hedonic value.

**Figure 11 brainsci-10-00921-f011:**
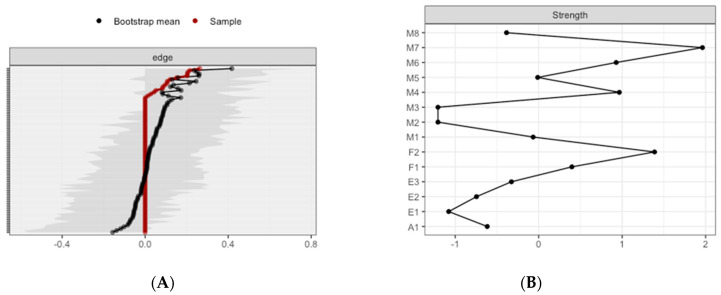
Estimated edge-weights for the estimated network and centrality of nodes. (**A**) Left: Bootstrapped CIs of estimated edge-weights for the estimated network. The red line indicates the sample values and the gray area the bootstrapped CIs. Each horizontal line represents one edge of the network, ordered from the edge with the highest edge-weight to the edge with the lowest edge-weight. (**B**) Right: Centrality of nodes within a network according to ETOC subsample.

**Table 1 brainsci-10-00921-t001:** Characteristics of participants for each Mindfulness Disposition (MD) cluster (mean ± standard deviation or sample size).

	MD Cluster on All Participants (*n* = 76)		MD Cluster on ETOC Subsample (*n* = 31)		
Sociodemographic Data	High MD	Low MD	High MD	Low MD	*p*-Value *
N	32	44	18	13	
Age	27 ± 5.59	30 ± 10.27	25 ± 4.91	30 ± 10.15	0.56
Woman/man	4/28	8/36	2/16	0/13	0.38
Submariners	17	11	8	6	
Marines	8	4	8	3	
Overwinterers	7	29	2	4	

* *p*-value for analysis of unit and Chi2 cluster effect calculated for age and gender. None of the measures had a significant cluster effect.

**Table 2 brainsci-10-00921-t002:** Summary of MD profile on psychological assessments (mean ± standard deviation).

Variables	High MD	Low MD	*p*-Value *
SPANE Positive	4.22 ± 0.47	3.94 ± 0.54	0.01
SPANE negative	2.43 ± 0.64	2.13 ± 0.54	0.02
MAIA not distracting	2.64 ± 0.76	2.96 ± 0.75	0.06
MAIA attention regulation	3.37 ± 0.68	2.51 ± 0.76	0.000
MAIA self-regulation	3.22 ± 0.79	2.48 ± 0.84	0.000
MAIA body listening	2.66 ± 1.18	1.93 ± 0.88	0.006
MAIA trusting	4.38 ± 0.59	3.55 ± 1.04	0.000
Subjective olfactory acuity	7.69 ± 1.97	6.27 ± 2.40	0.008
Subjective taste acuity	8.06 ± 1.79	7.70 ± 2.21	0.02
Subjective hearing acuity	8.97 ± 1.12	7.59 ± 2.17	0.005
Subjective acuity of balance	8.19 ± 1.45	7.11 ± 2.32	0.02
Subjective visual acuity	9.28 ± 0.92	8.59 ± 1.77	0.08

* *p*-value for analysis of unit group effect.

**Table 3 brainsci-10-00921-t003:** Summary of MD profile on objective olfactory assessments (mean ± standard deviation).

Variables	High MD	Low MD	*p*-Value *
ETOC identification	12.6 ± 1.50	11.3 ± 2.06	0.106

* *p*-value for analysis of unit group effect.

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
