# Peer review of "Mindfulness, Interoception, and Olfaction: A Network Approach"

_brainsci, 2020, doi:10.3390/brainsci10120921_

Round 1

Reviewer 1 Report

the study 'Mindfulness, interception and olfaction: a network approach' sounds really interesting. The authors investigated how Mindfulness disposition is associated with an interception/exteroception in olfactory cognition.

The methodology is correct, the study is really accurate and the results are really interesting (in particular for researcher in olfactory cognition field), and the limitations of the study are fully described 

Author Response

We would like to thank you for your feedback, and your interest to this study.

Reviewer 2 Report

This article presents exploratory data on the relationship between a disposition to mindfulness, interoceptive and exteroceptive reported perception and reported positive and negative feelings. Although I find the research question interesting and that the introduction is fairly well done, I think the data are too preliminary and the statistical power too low for the data and interpretation to be published as it is.

Major points:

  • The authors indicate that data collection is still ongoing (line 141), so I suggest that they continue to collect data in order to be able to draw more robust conclusions than the current ones. I am not aware of the context that prompted the authors to publish their data as it stands, but I think it would be much more interesting to increase the sample size to be able to properly explore the relationships between the variables and clearly answer the research question. However, I would not object to revising the article with the current data.
  • If the aim of the authors is to characterise the relationships between MD, interoception and emotion, why split into low and high MD groups? They should examine the partial correlation network on all the data in order to take full advantage of all the variability on the scales. In addition, the resulting network would certainly be much more robust because of the much larger sample size, which is always important when calculating correlations. I have the impression that the aim of the authors is rather to characterise the relationships between the variables, i.e., the partial correlation networks, for high and low MD, separately. Both questions are very interesting but the authors should clarify their position because their objective presented in lines 128 and 378 cannot be best achieved with the analysis strategy they have developed (separation into High vs Low MD).
  • In addition, some issues that I think are crucial are not addressed by the authors: Do the authors expect the same network with different weights according to the MD score or a re-organisation of the network from low to high MD? What are the implications of either hypothesis?

Other points:

Line 96 : « olfactory functionning » is too broad. What aspect of olfactory functioning is correlated with bodily signals?

Line 99: functions of olfaction, see also R.J. Stevenson (2010): An initial evaluation of the functions of human olfaction, Chem. Sens. 35, 3–20.

Line 99: what is a primary sense? Do we have secondary senses? Please define or remove.

Line 105: see also E.A. Krusemark, L.R. Novak, D.R. Gitelman, W. Li (2013). When the sense of smellmeets emotion: Anxietystate- dependent olfactory processing and neural circuitry adaptation, J. Neurosci. 25, 15324–15332.

Line 112: I do not see the point of introducing, even so succinctly, the literature on the Event Related Potentials. The authors do not extract any relevant information for their arguments. I think that this small paragraph can simply be deleted.

Line 168: isn’t it rather “Emotional life or functioning” than psychological functioning?

Line 172: please provide the different items of the questionnaire in supplementary material. Without this, it is impossible for the reader/researcher figure out what it effectively measured.

Line 185:

Line 193: Please be more specific, which kind of data? All variables? Which mean? What is the rationale for that?

Line 196: It is appreciable that the authors have verified the normality of the distributions as well as the homogeneity of the variances. Mention at this level that the statistics are adapted to each situation and the t or W is then reported in the results.

Line 197: I think it's really important that the authors report the effect sizes as well as their confidence intervals, not just the p-value which is not informative on the size/strength of the effect they observed. See for instance: Lakens D (2013) Calculating and reporting effect sizes to facilitate cumulative science: a practical primer for t-tests and ANOVAs. Front. Psychol. 4:863. doi: 10.3389/fpsyg.2013.00863. Similar procedures exist for non parametric estimates.

Line 201: The authors introduce the notion of causal model, causal interaction in the introduction and then network interaction in the methodology. If all these terms cover different procedures then much more precision is needed. On the other hand, if these terms cover the same procedure, then a coherent vocabulary must be adopted throughout the article.

More generally, I think that the authors have to take precautions with the notion of causality. Even if it is very clear to them, they should be very careful not to imply that causal links are demonstrated by these analytical techniques. It is only suggested, assuming that there is no hidden latent unmeasured variable. As mentioned by Epskamp and Fried, 2018): “…partial correlation networks are thought of as highly exploratory hypothesis-generating structures, indicative of potential causal effects.” Epskamp, S., & Fried, E. I. (2018). A tutorial on regularized partial correlation networks. Psychological methods, 23(4), 617.

Line 236 : incorrect wording, please rephrase.

Line 241: There is no need to report all the variables’ means both in the text and in Table II. It is preferable to report those values and statistics in the table (adding the effect sizes and their confidence intervals) and to summarize the message in the text.

Line 279: I think that by reporting the effect sizes, the authors will no longer be able to interpret this trend. More generally, an interesting approach could be to calculate a Bayes factor in order to determine if the absence of effect is due to a small sample size or if H0 is really more likely (https://jasp-stats.org/).

Line 289: The abbreviations S1, S2 etc. do not appear in the figure.

Line 397: for MD group…. Which group? Also line 407

Line 430: “to feel odours” do you mean to smell odours or the ability to express an odour related feeling?

There is no conclusion after the limitations?

Author Response

The authors indicate that data collection is still ongoing (line 141), so I suggest that they continue to collect data in order to be able to draw more robust conclusions than the current ones. I am not aware of the context that prompted the authors to publish their data as it stands, but I think it would be much more interesting to increase the sample size to be able to properly explore the relationships between the variables and clearly answer the research question. However, I would not object to revising the article with the current data.

Correction: The Covid-19 pandemic stopped data collection, for the next two years all studies have been cancelled. Furthermore, I wanted to specify that the data collection concerning the same participants. One of the corollaries of doing research in ICE/EU environments is the small sample size. We need to be flexible and adaptable to these ecological environments where professionals feel the psychological and physiological constraints of isolation and confinement. However, this is also the strength of these studies. By focusing on these populations, we allow a better consideration of the needs in isolated and confined environments, which with the Covid-19 pandemic can have real benefits in addition to improving mission conditions. 

If the aim of the authors is to characterise the relationships between MD, interoception and emotion, why split into low and high MD groups? They should examine the partial correlation network on all the data in order to take full advantage of all the variability on the scales. In addition, the resulting network would certainly be much more robust because of the much larger sample size, which is always important when calculating correlations. I have the impression that the aim of the authors is rather to characterise the relationships between the variables, i.e., the partial correlation networks, for high and low MD, separately. Both questions are very interesting but the authors should clarify their position because their objective presented in lines 128 and 378 cannot be best achieved with the analysis strategy they have developed (separation into High vs Low MD).

Correction: The aim of this study is to explore the relations between mindfulness, interoception, emotions and exteroception. The literature has shown that mindful subjects, depending on their level, had a higher or lower interception, hence the relevance of separating our population. By taking the whole of our cohort, we therefore overwrite the group effects which allow us to highlight relationships according to the mindful profile of the participants. However, we have added in the "results" section, two interaction networks on all subjects (n=76) from lines 240 to 343 in order to explore the relationship between interoception, emotions and subjective olfactory acuity, then after this characterization between mindfulness, interoception, emotions, and subjective extrasensors acuity. Thus, we make the hypothesis that reciprocal relationships exist between them. The second issue will part in a categorical approach focusing on whether the MD is associated with differential interoception, exteroception and emotion. Our hypothesis is that subjects with MD exhibited higher level of interoception, exteroception including olfaction and emotion. The third hypothesis aims to evaluate how MD interacts with interoception, olfaction and emotion using a causal network approach. We assume that network causal relationships differ according to the MD profile, with more connections among interoception, olfaction and positive emotion for high MD groups (see line to line).

In addition, some issues that I think are crucial are not addressed by the authors: Do the authors expect the same network with different weights according to the MD score or a re-organisation of the network from low to high MD? What are the implications of either hypothesis?

Correction: We expected differences in interoception, olfaction and emotion interactions’ between low MD networks compared to high MD networks. Our results confirmed this hypothesis. While in the low MD group we have interactions only between interoceptive variables, in the high MD group we have more interaction between interoceptive and positive emotions variables. These results highlight and confirm that the level of MD influences the subjects’ ability to feel and perceive the world with potential implications for their ability to adapt to ICE/EUE environments.

Line 96: « olfactory functionning » is too broad. What aspect of olfactory functioning is correlated with bodily signals?

Correction: Corrected line 97: « (detection threshold, odor discrimination, and odor identification) ».

Line 99: functions of olfaction, see also R.J. Stevenson (2010): An initial evaluation of the functions of human olfaction, Chem. Sens. 35, 3–20.

Correction: Thank you for the reference.

Line 99: what is a primary sense? Do we have secondary senses? Please define or remove.

Correction: Corrected line 101: « (i.e., sight, hearing, smell, taste, touch) ».

Line 105: see also E.A. Krusemark, L.R. Novak, D.R. Gitelman, W. Li (2013). When the sense of smellmeets emotion: Anxietystate- dependent olfactory processing and neural circuitry adaptation, J. Neurosci. 25, 15324–15332.

Correction: Thank you for the reference.

Line 112: I do not see the point of introducing, even so succinctly, the literature on the Event Related Potentials. The authors do not extract any relevant information for their arguments. I think that this small paragraph can simply be deleted.

Correction: Corrected line 115: « for better understanding how mindfulness works ». We briefly introduce this notion because it is within the framework of the work of ERPs that a better understanding of the mindful disposition has been made.

Line 168: isn’t it rather “Emotional life or functioning” than psychological functioning?

Correction: Corrected line 173 « Emotional functioning ».

Line 172: please provide the different items of the questionnaire in supplementary material. Without this, it is impossible for the reader/researcher figure out what it effectively measured.

Correction: The questionnaires used in this study were reported in supplementary material. However, they were in French. The referenced articles provide an English version.

Line 185:

Correction: This section is empty.

Line 193: Please be more specific, which kind of data? All variables? Which mean? What is the rationale for that?

Correction: Corrected line 198: « FMI data ». More precisely, FMI data include both factors « presence » and « Non-judgmental acceptance ». The StandardScaler function, from the scikit-learn package in Python, was used in order to standardize the data by removing the mean and scaling to unit variance. For more explications, see the following link (https://scikit-learn.org/stable/modules/generated/sklearn.preprocessing.StandardScaler.html).

Line 196: It is appreciable that the authors have verified the normality of the distributions as well as the homogeneity of the variances. Mention at this level that the statistics are adapted to each situation and the t or W is then reported in the results.

Correction: Corrected line 202 to 203: « The statistics were adapted following the results of the previous tests ».

Line 197: I think it's really important that the authors report the effect sizes as well as their confidence intervals, not just the p-value which is not informative on the size/strength of the effect they observed. See for instance: Lakens D (2013) Calculating and reporting effect sizes to facilitate cumulative science: a practical primer for t-tests and ANOVAs. Front. Psychol. 4:863. doi: 10.3389/fpsyg.2013.00863. Similar procedures exist for non parametric estimates.

Correction: Corrected line 206: « For significant analyses, the effect sizes and their confidence intervals were reported ».

Line 201: The authors introduce the notion of causal model, causal interaction in the introduction and then network interaction in the methodology. If all these terms cover different procedures then much more precision is needed. On the other hand, if these terms cover the same procedure, then a coherent vocabulary must be adopted throughout the article. More generally, I think that the authors have to take precautions with the notion of causality. Even if it is very clear to them, they should be very careful not to imply that causal links are demonstrated by these analytical techniques. It is only suggested, assuming that there is no hidden latent unmeasured variable. As mentioned by Epskamp and Fried, 2018): “…partial correlation networks are thought of as highly exploratory hypothesis-generating structures, indicative of potential causal effects.” Epskamp, S., & Fried, E. I. (2018). A tutorial on regularized partial correlation networks. Psychological methods, 23(4), 617.

Correction: Thank you for your advice. In order to have a more coherent vocabulary, we used « causal interaction network », and « presumed » line 126. The exploration of connections between system variables is relevant when there is a silence in the literature on the topic.

Line 236 : incorrect wording, please rephrase.

Correction: Corrected line 353 : « a higher MD by ».

Line 241: There is no need to report all the variables’ means both in the text and in Table II. It is preferable to report those values and statistics in the table (adding the effect sizes and their confidence intervals) and to summarize the message in the text.

Correction : Thank for your advice, the changes were reported in the results.

Line 279: I think that by reporting the effect sizes, the authors will no longer be able to interpret this trend. More generally, an interesting approach could be to calculate a Bayes factor in order to determine if the absence of effect is due to a small sample size or if H0 is really more likely (https://jasp-stats.org/).

Correction: In calculating the size of the effects, we report moderate to large effects, including trend. We did not test the likelihood of a difference, which may be an additional list to our exploratory study.

Line 289: The abbreviations S1, S2 etc. do not appear in the figure.

Correction: The different labels were corrected in the figure of both MD profiles (Figure 7).

Line 397: for MD group…. Which group? Also line 407

Correction: Corrected line 570 « the high MD group » and line 410 « high MD subjects ».

Line 430: “to feel odours” do you mean to smell odours or the ability to express an odour related feeling?

Correction: Corrected line 602 : « to smell odors ».

There is no conclusion after the limitations?

Correction: A conclusion was added after the limitations of the study lines 644 to 656.

Round 2

Reviewer 2 Report

The authors have taken the remarks into account and after some minor modifications,  However, I think the text would benefit greatly from a proofreading by an English speaker. The text is understandable but many of the formulations are awkward. 

Lines 94-95: The authors' sentence is still unclear. Currently it means that as cardiac interoceptive accuracy increases, detection thresholds decrease, discrimination decreases and identification decreases. If the latter two reflect a regression in olfactory capacity, the lower the detection thresholds, the better the olfactory capacity. Thus the three measures cited are in contradiction. The authors must therefore be more precise.

Line 99: Why not simply mention "one of our 5 senses". The notion of primary meaning is not relevant.

Line 113: “to better understand how mindfulness works.”

Line 357 and followings: I appreciate the inclusion of effect sizes and confidence intervals. However, the minimum of the interval should be on the left and the maximum on the right. This is currently not the case all the time.